# Identification and Classification of the Tea Samples by Using Sensory Mechanism and Arduino UNO

**Amruta Patil** [1], **Mrinal Bachute** [1,*] and **Ketan Kotecha** [2,*]

1. Electronics and Telecommunication Engineering Department, Symbiosis Institute of Technology (SIT), Symbiosis International (Deemed University), Pune 412115, India; amruta.patil.phd2019@sitpune.edu.in
2. Symbiosis Centre for Applied Artificial Intelligence, Symbiosis International (Deemed University), Pune 412115, India
* Correspondence: mrinal.bachute@sitpune.edu.in (M.B.); head@scaai.siu.edu.in (K.K.)

**Abstract:** Tea is the most popular hot beverageworldwide. In 2020, the value of the global tea market was almost USD 200 billion, and is estimated to reach up to USD 318 billion by the year 2025. Tea has been included as part ofa regular diet for centuries because of its various health benefits. However, tea is acidic, and over-consumption causes heat problems, disturbance of the sleep cycle, tooth erosion, and low calcium absorption in the body. Strong tea concentration is very harmful and toxic. The safe consumption of tea should be guaranteed. The treatment applied in this research work is on sensory mechanisms and Arduino UNO. The objective of this paper is to find out community interest in a particular tea species and inform them about tea overdose.The acidity is mapped with tea taste in terms of strong, medium, and low flavors. Based on the data analysis, the results differentiatethe acidity level of black tea (pH: 3.89–4.08) as very high, green tea (pH: 4.68–4.70) is in the 2nd position, and the energy drink Herbalife Nutrition (pH: 5.59–5.64) is the least acidic comparatively, with a proportion ratio 1:10 of tea to water. Experimental analysis reveals that in the additives, lemon is most acidic, followed byginger, lemongrass, and Tulasi.

**Keywords:** artificial intelligence (AI); artificial taste perception; machine learning (ML); pH measurement; sensory mechanism; tea samples

## 1. Introduction

India is the second-largest tea-producing country in the world. Various Indian tea brands such as Assam tea and Darjeeling tea are famous for their unique tastes and flavors. In agricultural countries such asIndia and China, tea is an economic crop and asset for the country. Therefore, the marketing of a tea is dependent mainly on its attributes such asflavor, fragrance, and color [1–6]. Nowadays, the major worry is economically triggered malpractices in food and beverages. Only safety norms cannot control such activities. High-end equipment cannot be purchased everywhere in the trade because of cost. Low-cost handheld devices can give a transparent view of food and beverages. Thiswill assure the safety and quality of beverages anytime, anywhere, by the seller and customer. This kind of micro-planning can suppress cheap practices such ascontamination and adulteration from agricultural product sales.

For centuries in India, the practice of medicines by herbs, plants, and natural resources, named Ayurveda, has prevailed. Tea is herbal medicine, offers many benefits, and is hence widely consumed and included as a part of a daily routine.

The tea board of India regulates tea-related activities [7]. The farming, processing, and marketing strategies play an important role in the tea business. These days, the flavor, color, and fragranceadulteration in teais a matter of concern [7].Standard norms have been provided by the Tea Board of India to all the tea manufacturers and sellers to warn about tea alteration for color, flavor, and fragrance [7]. In India, synthetic color for tea has been banned by the Tea Board. Most of the time, damaged tea leaves had been improved

with various coloring agents for their appearance in order to obtain a good price. The same procedure of dyeing is carried outwith an old tea stock. Prussian blue, turmeric, andindigo are common dyes used for adulteration, improvingcolor, or glossiness in tea that is non-toxic for humans [7]. However, the refreshing effect of the flavor differs due to nontoxic adulteration. Synthetic color agents are harmful and cause other side effects. Figure 1 shows the tea flavor grading based on its pH value.

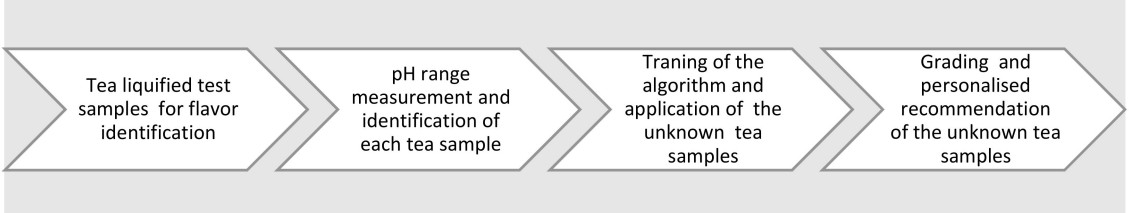

**Figure 1.** Tea flavor grading based on its pH value.

For many years in India, tea tasters were appointed for taste-based grading of the tea. A longer job period creates taste saturation and psychological effects on tasters and the tea taste grading. It leads to biased and inconsistent responses after some duration due to loss of taste sensitivity.

Chemical tests and equipment such ashigh-performance liquid chromatography (HPLC) and gaschromatography (GC) are available. The equipment detects the proportion of dissolved compounds and does not show any mapping in terms of taste, odor, and color. A skilled person is required to operate the test equipment. The HPLC and GC equipment are around INR 8 to INR 10 lakh. The HPLC and GC equipment cost as well as the maintenance cost are very high.The chemical tests are time-consuming and equipment is not handy. The monitoring and control in tea manufacturing and tasting is notpossible with a manual system. To manage large-scale tea production, artificial intelligence is very much obligatory.

The purpose of the work is to identify and control tea adulterationfor buyers and consumers using artificial intelligence. It can help to find the age of tea stock by observing the pH for flavor and color appearance change by using sensors.AI is valuable to create the standard dataset of tea attributes for various tea species, as well is useful to compare the newly acquired sample with the standard dataset. The AI-based solutions are comparatively cheaper and can generate direct mapping with the physical attributes of the tea. Simple and handy designs are possible with AI and any unskilled person can easily operate such devices.

In an artificial taste perception, the abilityof the tongue to analyze and segregate taste patterns of various tea species can be artificially formed using sensors and software algorithms. The programmed hardware will be able to discriminate various tea tastes and the taste patterns of a single sample with different concentrations. In this research article, the pH sensor and Arduino UNO have been used for testing the tea samples and additives.

Artificial perception is useful in giving preventive advice for reducing consumption of acidic beverages and recommend safe quantities and alternatives. Black tea is slightly acidic and can increase acidity quickly after consumption on the tooth's surface. The presence of fluoride in black tea improves oral health [1]. However, the sugar content and acidity of the drink can cause dental caries and erosion. A long and fast working culture in today's world leads to unhealthy habits such as overconsumption of tea. Unfortunately, the drinks are heavily advertised without proper guidelines and personalized recommendations. Tea is consumed for its antioxidant properties, which bring healthy immunity. Tea consumption is very high in India, as it improves brain functionality. So-called 'workaholics'consume tea many times a day to avoid sleep.

It is permittable to consume one's desiredquantity of tea and coffee on a daily basis, but overconsumption may disturb the sleep cycle, heart problems, dental erosion, and

acidity [8,9]. In some diseases such as diabetes and thyroid disease, consumers should know the dose of tea and coffee in order to control and avoid health consequences [8]. The authors of the paper [8] had discussed the impact of the addition of sugar, lemon, and milk in the tea. The addition of sugar in tea can cause dental caries, whereas the addition of lemon with the prescribed quantity is safe. The addition of milk is desirable as it is not affecting the pH value of tea. The fact of tea consumption with a meal has been discussed and concluded that tea consumption reduces the absorption of dietary iron. The possibility of low iron absorption creates harmful body reactions in the long run. Tea is not prescribed for young children because of the above-said fact. Tea consumption has been analyzed by experimentation on the ten people aged 21–23 for pH effect on teeth and oral health. The tool used is a micro-pH electrode which is mounted on a vinyl splint [8]. The pH of tea detected initially was 4.9 andmostlycontains oxalate and nitrate anions. Some various classes and objectives which need tea prescription have been shown in Figure 2.

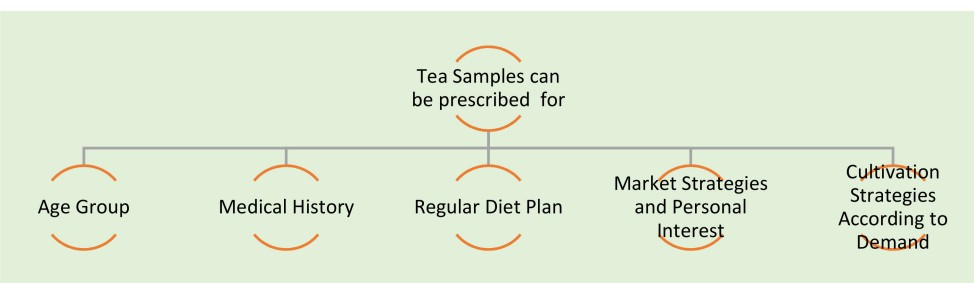

**Figure 2.** Tea prescription for various classes and their objectives.

The effect of pH on green tea extraction isexplained in the paper [10]. Green tea infusions were prepared and analyzed with different pH liquid samples. The color of green tea extraction was increased with the pH value increment. Total amino acid contents in green tea were not differed significantly by pH value. This paper illustrates the effect of pH on the total contents of catechin of the green tea extraction. Catechin decreases with the increment in pH value. The increment in pH value causes an increment in caffeine contents [10,11]. Therefore, a low pH value is preferred for green tea preparation [10].

Green tea can be used in mouthwashes, toothpaste, and dentifrices because of its biochemical properties [12,13]. Tea is the mixture of the leaves of the Camellia sinensis plant [12]. The pH value of drinks and beverages causes dental erosion without the presence of bacteria. The low value of pH indicates the acidity of the drink. These drinks cause the chemical dissolution of teeth that results in dental cavities and decay [14]. The frequent consumption of high acidic or low pH drinks is harmful to all ages in human beings [14].

The study of the effect of various brewing methods on the quality of green tea was discussed in the paper [15–19]. The study informs the brewing method affects the color, aroma [20,21], and taste of green tea. In addition, the study considers soluble solids, pH value, catechins profiles, and caffeine and antioxidant components. The tea infusions for testing were made with cold, hot, and steam processes at 2, 6, and 10%. Cold brewing of tea results in mild taste and color, whereas hot and steamed tea infusions have a dark color and strong flavor [15,22–24]. Figure 3 shows the three popular varieties of tea, of which black tea was most popular for centuries, and cultivation of it spread everywhere in the world [21].

The polyphenols found in black tea have higher antioxidant properties. Green tea is a semi-fermented type of tea, and it keeps a high body metabolism [15,25]. The unfermented tea is Oolong tea. It is rarely cultivated, and it supports good digestion [15]. Green tea has prepared in two styles-one by steaming and is popular in Japan. The second style is famous in China for parched green tea preparation [15–18,26,27]. Preparation of tea infusions is carried out with steam and cold methods with 2, 6, and 10% ground leaf tea powder (70, 210, 350 g) with 3500 mL of water. In cold style, cold water (4C) has been

used with a shaky bath (SB302, Kansan Instruments Co., Kaohsiung, Taiwan) at a speed of 100 rpm for 24 h. In the steam case, tea powder mixed with hot water (90C) and a shaky bath has been applied at a speed of 100 rpm for 20 min. Both style samples were filtered and cooled to room temperature for testing. The pH value, total dissolved solids(TDS), and degree of Brix of both cases have been measured with three different equipment Sp-701 pH meter (Suntex Instruments Inc., Taipei City, Taiwan), a JENCO 113 *conductivity* meter (Jenco Electronics, New Taipei City, Taiwan), and an N-1a hand refractometer (Atago Co., Tokyo, Japan), respectively [15–18]. Figure 4 describes the tea parameters such as the potential of hydrogen ion concentration (pH), total dissolved solids (TDS), and Brix for dissolved sugars were electrically transformed for the tea liquor analysis.

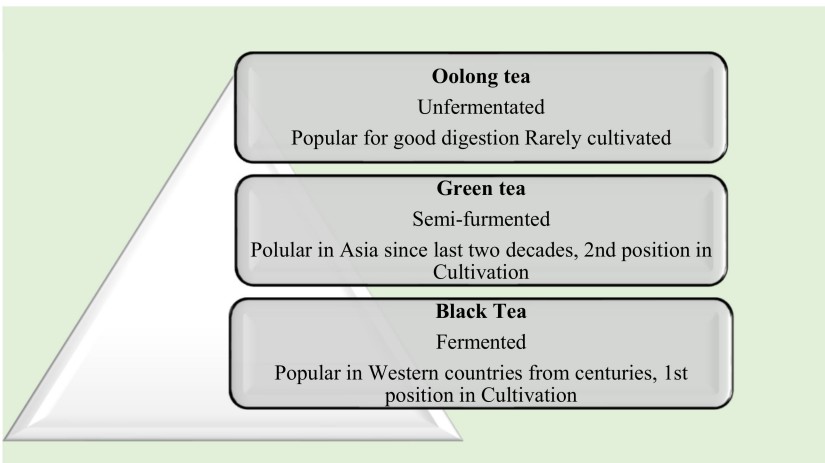

**Figure 3.** Tea types and their demands.

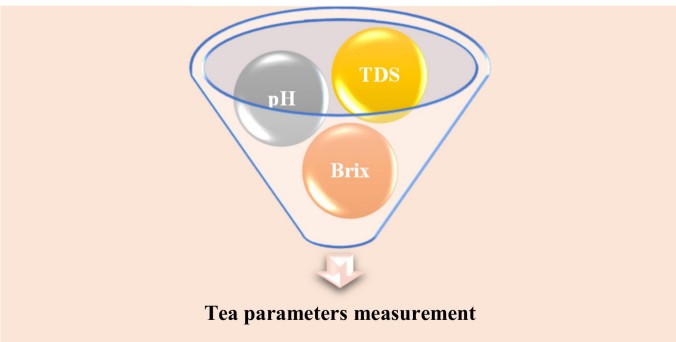

**Figure 4.** Electrically transformed tea parameters.

The pooled analysis of the pH, TDS, and Brix is important. The pH level is useful to find the acidity of the aqueous sample.

The TDS describes total dissolved solids in terms of conductivity or the current value. The degree of Brix is specifically for the sugar content of the liquid. One degree of Brix is equivalent to 1 g of sucrose in 100 g of solution.

For color measurement, three values, 'L', 'a', and 'b', were measured with the S80 Color Measuring System (Nippon Denshoku Inc., Tokyo, Japan). The L values range from 0 to 100 and are mapped dark to the lighter color reflection of tea infusion. The positive and negative 'a' values have been mapped for the redness to green-ness, whereas positive and negative 'b' values define yellowness and blueness [27]. An effect of brewing methods is moreover the same on pH values. The parched green tea pH value was observed in the range 5.47–5.66, and for the steam, green tea pH is observed in the range 5.69–5.88. TDS and Brix values observed were higher for high concentrations (high % of tea leaves to water). Cold brewing TDS and Brix values are comparatively lower than steamed brewing [27]. There is not much effective color difference observed due to different types of brewing.

The tea infusions for cold and steamed style with 2% concentration are ready to drink but should be diluted by 6% and 10% for both styles before drinking [27,28].

The secondary compounds Saponins in tea form stable foam in liquid solutions, which possess antimicrobial, antivirus, anti-inflammatory, antioxidant, antiallergic properties. These are widely used in various health care products and medicines. The effect of pH on tea Saponins was observed in the paper [29]. For the pH value 8, Saponins shows the best antioxidant properties, whereas pH value 4.8 indicates higher antibacterial activities [29]. The effect of tea consumption on salivary pH has been explained in the paper [30] for sugar-free liquor and various other additives such as sugar, jaggery, and stevia powder. The effect on the pH of saliva was observed after 1 min, 20 min, and 60 min. It is faster for stevia powder to regain the initial pH before drinking than sugar and jaggery, and itmatcheswith plain tea drinks [30].

The paper study [31] describes that the stability of tea polyphenols depends on pH and temperature value. The effect of low pH and high fluoride content detection in drinks has been explained in detail in the Study [32–37]. To control the pH and fluoride content of the drinks is recommended to prevent dental fluorosis and erosion [32]. The method stated in the paper is using a digital pH meter and digital fluoride meter. For the tea, the sample pH is 5.18 ± 0.05 [32,33,37,38]. The paper [33] describes the motivation behind tea, coffee, and alcohol consumption. The finding shows the relation between bitter taste and tea consumption. In the case of coffee, when the intensity of quinine was more, consumption may reduce, and when the perception of caffeine was higher leads to coffee consumption. The effect in the case of tea consumption was the opposite. Similarly, the way study [39] says caffeine and non-galloylated Catechin causes' bitter taste of tea. The presence of compounds—flavonol-O-glycosides, tannins, and allocated catechins generates astringent taste in tea infusion [39].

Few principal conclusions derived from the main work area are as follows- conventional tea testing equipment such as liquid chromatography (LC), gas chromatography (GC) [40] are not handy, time-consuming, costly, and need a skilled operator. Since tea cultivation in India, tea testing is manual, and skilled human tea tasters decide the ultimate tea taste grade. Sometimes physiological effects and long-time work may lead to taste saturation in manual tests by tea tasters and may create biased results. The basic taste of tea is bitter and astringent, but it is a non-alcoholic drink [3,4,27,41,42].On the other hand, the refreshing feel of green tea is due to the umami taste [41,43–65].

The term "pH" stands for potential hydrogen (H+) ion inan aqueous solution. The pH level decides the acidity, neutrality, and alkalinity of any liquid. The range of pH varies from 0–14. The substances that belong to the pH value below seven are acidic. The pH value 7 indicates the substance is neutral, whereas the pH value above 7 specifies the alkaline substance category. Figure 5 shows the pH scale for the nature of the liquid.

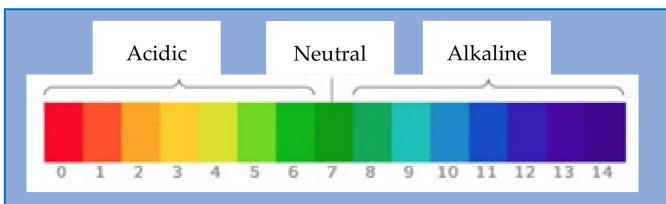

**Figure 5.** The pH scale.

The "(1)" and "(2)" indicate the pH value expressions

$$pH = -\log [H^+] \tag{1}$$

$$pH = \log (1/[H^+]) \tag{2}$$

An acid is the hydrogen ion donor. If the solution contains more hydrogen ions than the hydroxide ions, then it is acidic. The pH value of acidic solution near 7 indicates weaker

acid, whereas a value far from 7 indicates it is a stronger acid. If the proportion of the hydroxide ions is more than the hydrogen ions, the solution is base or alkaline in nature. The value nearer to 7 shows a weaker alkaline solution for a base solution, whereas the pH value far away from 7 indicates a strongly alkaline solution. The solution made up of an equal concentration of hydrogen and hydroxide ions is neutral. For the pure water ph value is 7. Nowadays, digital pH meters are very popular for pH measurement as they are fast and accurate.

Normally, tea is acidic, but the acidity may increase or decrease with the substances we add to it for the flavor variety. The harmless ph value of the tea is 5.5. General pH values of various tea types are shown in Table 1.

**Table 1.** General pH values for main tea samples.

| Tea Type | pH Value |
| --- | --- |
| Black Tea | 4.9–5.5 |
| Green Tea | 7–10 |
| Oolong Tea | 5.9 to 8.2 |
| Lemon Tea | 3 |

## 2. Materials and Methods

The primary requirements of the artificial taste perception setup have been shown in Figure 6. At the receptor level, the reception mechanism in which the taste has been sensed through the tongue's taste buds. The sensing property of the tongue to analyze taste patterns for bitterness can be artificially formed with the taste sensors. First, the taste sensor senses the flavor of liquor and converts it into an analog signal. The next level is the transmission. In this step, the perceived taste stimuli have been transmitted to the brain through nerves in natural systems. In a synthetic system, the sensor's analog output has been converted into digital form and then transmitted to the processor unit through a signal conditioning and data acquisition module [21,40].

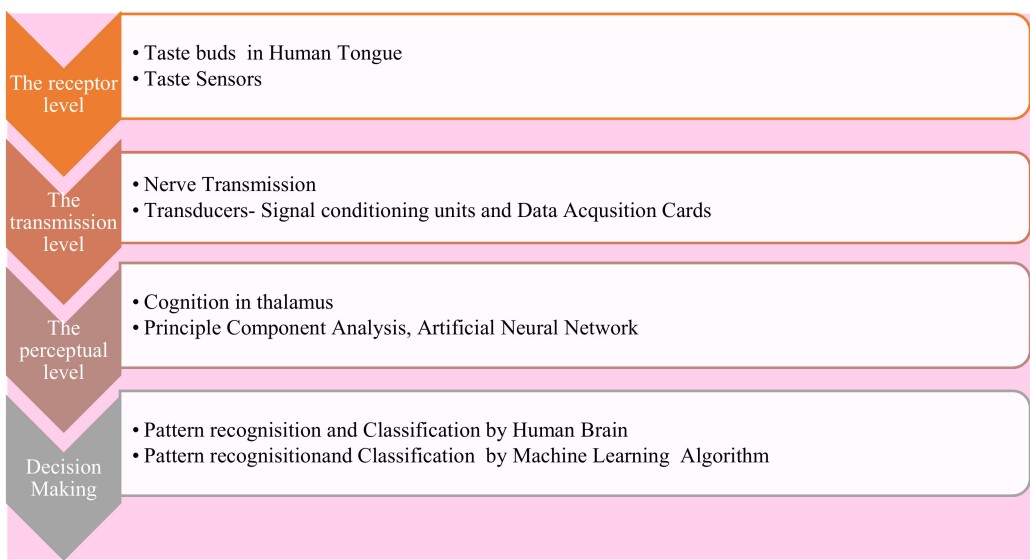

**Figure 6.** Artificial taste perception.

The cognition in the thalamus happens at the perceptual level. With the help of the principal component analysis (PCA) and artificial neural network (ANN), artificial processing can form. Finally, the decision of the type of flavor has been taken by the brain. In any synthetic system, the logical process of flavor decisions has been made with the help of machine learning [21,40].

## 2.1. Research Problem

To form the standard dataset for the tea flavor testing and grading by using sensory mechanism and Arduino UNO.

## 2.2. Research Objective of This Study

Development of an electronic gadget:

1. Thatwill classify the tea samples—and recommend the tea for the specific health diet based on its pH level.
2. Thatwill find the effects of additives such as lemon, ginger, lemongrass, etc. on tea pH value.
3. It will give early intimation for the quantity that tea lovers can consume.

Figure 7 shows the block schematic of the sensing mechanism of a tea sample using an Arduino UNO. It includes a tea sample pot, a pH sensor with a BNC probe and signal conditioning module, and an Arduino UNO processing board.

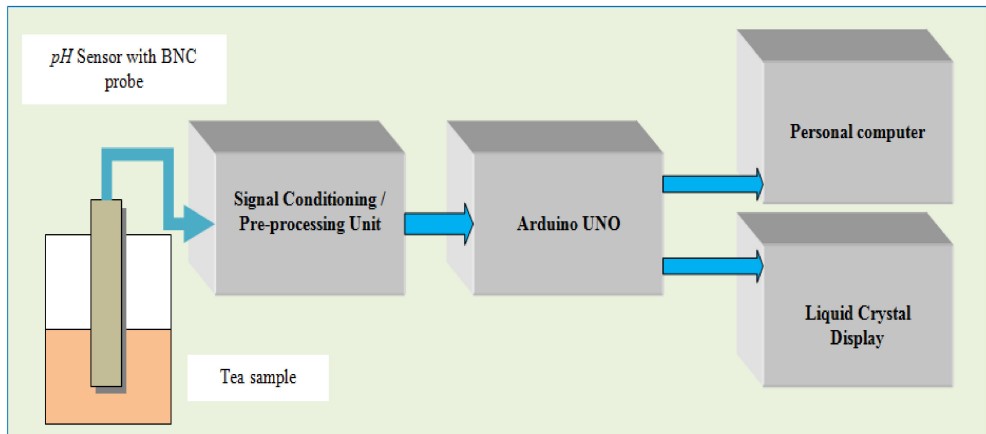

**Figure 7.** Sensing mechanism of tea sample using Arduino UNO.

## 2.3. The pH Sensor and Its Principle

The analog pH sensor module from DIY MORE has been used for pH measurement. The ph sensor module consists of two parts, sensor with BNC probe and signal conditioning or preamplifier board.

### 2.3.1. The pH Sensor Plastic Tube

Figure 8 gives the view of the physical appearance of the ph sensor. The single sensor consists of a working and reference electrode. The electrodes are made up of silver wire and coated with silver chloride. The reference solution is saturated potassium chloride (KCL) solution filled inside a plastic tube surrounded by a reference electrode. The working electrode is immersed into the glass tube ending with a glass bulb. The glass tube has filled with hydrogen chloride (HCL) solution. By measuring the potential difference between electrodes, the pH value can be measured.

When immersed into the solution, a ph sensor, the H+ ions sensitive glass membrane attracts the H+ ions from that solution. Internally, the H+ ions attract towards the glass membrane at the bulb, which will cause a potential difference between electrodes. For example, if the hydrogen ion concentration outside the glass bulb is more than inside, then the solution is acidic, whereas it is alkaline for the reverse case. The signal conversion module will measure the potential difference.

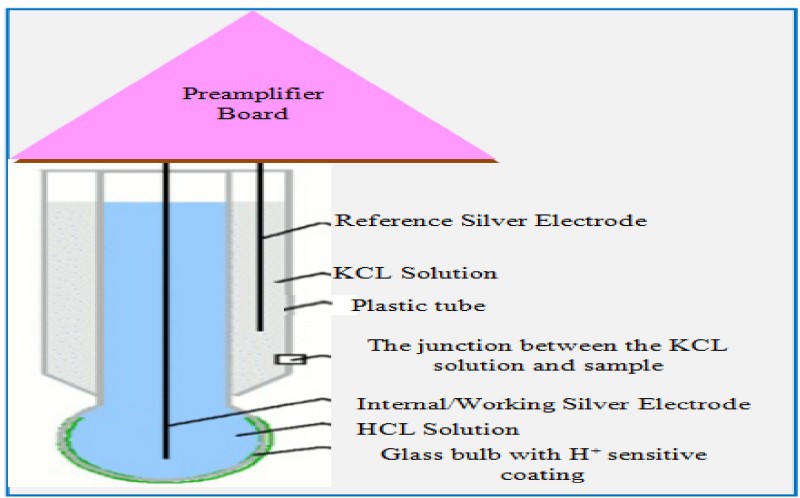

**Figure 8.** The ph sensor.

2.3.2. Signal Conditioning and Pre-Processing Unit

The pre-processing unit is given in Figure 9. The potential difference acquired from electrodes has been given to signal conversion or pre-processing units. The potential of the working electrode is indicated by VpH, whereas Vref indicates the potential of the reference electrode. The potential difference between the working electrode (WE) and the reference electrode (RE) is indicated by V as shown in below "3".

$$V = VpH - Vref \tag{3}$$

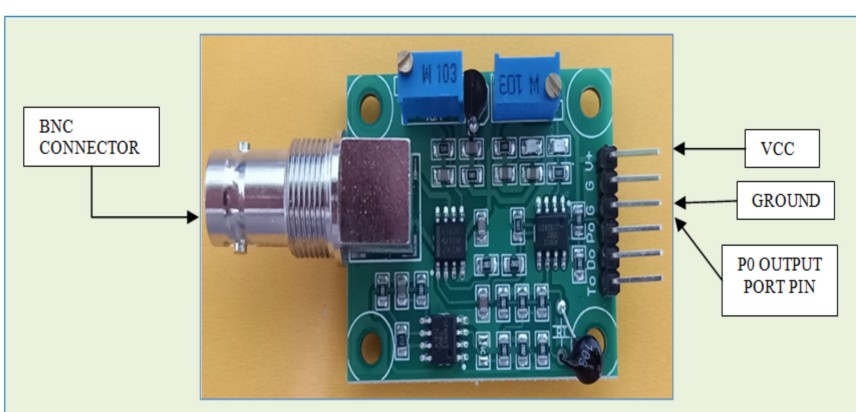

**Figure 9.** The ph sensor breakthrough board.

The potential Difference V is calculated by the breakthrough module of the analog pH sensor. The breakthrough module will be pre-amplified and mapped with the actual pH value of the liquid sample. The signal conversion (to find and map ph value) is based on the Nernst Equation (4) given below.

$$V = -kTpH \tag{4}$$

In Equation (4), k is Boltzmann's Constant, and T is the temperature. Table 2 shows the specifications of the pH sensor. The operating voltage is about 5 V. The working current is between 5 to 10 mA, and the module will detect the whole pH range from 0 to 14. Mapping of the output port voltage and the pH value at calibration is given in Table 3. For example, the acidic sample mentions calibrating the sensor to 3.031 V for the pH value 4; calibration for neutral is about 2.535 V. For the alkaline solution, the adjusted port voltage is 2.066 for a pH value of 10.

**Table 2.** Specifications of the pH sensor.

| Parameter | Limiting Values |
|---|---|
| Input supply voltage | 5 V |
| Working current | 5–10 mA |
| pH detection range | 0–14 |
| Temperature detection range | 0–80 °C |
| Response time | ≤5 s |
| Stability time | ≤The 60 s |
| Output | Analog |
| Power consumption | ≤0.5 W |
| Working temperature | −10 to +50 °C |
| Working humidity | 95%RH(nominal humidity 65%RH) |

**Table 3.** pH value and output voltage relevance.

| pH Value | Output (V) |
|---|---|
| 4 | 3.071 |
| 7 | 2.535 |
| 10 | 2.066 |

### 2.3.3. Arduino UNO

Arduino ATMega328P board is a hardware device that consists of 14 digital input-output pins. Out of the 14 I/O pins, six pins are available for the use of PWM output. It also consists of 6 analog Inputs. There is a 16 MHz quartz crystal, reset button, ICSP header, and a power jack. Simply connecting Arduino with a USB cable to a laptop or personal computer or by using an AC-to-DC adapter or with a battery, we can power it up. It is a low-power CMOS 8-bit microcontroller having enhanced RISC (Reduced Instruction Set Computer). To increase the performance of the Arduino, the AVR usesHarvard architecture. Figure 10 shows the Arduino implementation board [37,66–69].

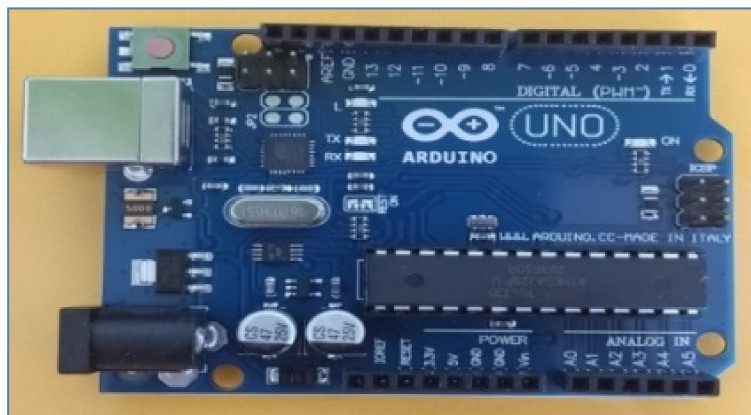

**Figure 10.** Arduino UNO implementation board.

### 2.3.4. Arduino Window-Based Software (IDE)

The Arduino Integrated Development Environment is the software used to write and upload programming codes to Arduino compatible boards. The separate versions of IDE are available for Windows, macOS, and Linux. The functions used in IDE are the same as that of C and C++. It is open-source software. The code written in IDE is known as a sketch. IDE does not need an external programmer. IDE software operates on any

operating system, and the programming is easy in IDE. IDE displays port output on the terminal for the visualization of value [68,69].

### *2.4. Algorithm for pH Sensor Calibration and Measurement Sensor Calibration*

#### 2.4.1. pH Sensor Calibration

Calibration is the process of correction in output values of circuits and devices by adjusting the potentiometer or trimmer values to obtain the standard known output for known input. Calibration is required for the sensors before actual measurement. The known values of the pH sensor as mentioned in Table 3 to calibrate pH values. The breakthrough board BNC connector external and internal parts need to be short-circuited with thecopper wire. It will force the 2.5 volts on the analog port pin P0 of the breakthrough board. The 2.5 Volts is obtained for pH value 7, which indicates a neutral input. Then, V+ pin of the breakthrough board is connected to Arduino's 5V pin, the P0 pin is connected to Arduino's Analog port pin A0, and the ground (GND) pin of Arduino and breakthrough board is short-circuited. Arduino is connected to the computer through a universal serial bus (USB) port, and the calibration code is uploaded to it through the integrated development environment(IDE) software.

#### 2.4.2. pH Sensor Measurement

Figure 11 describes the pH measurement algorithm. The variables were declared for the calibration value and the port voltage. The ten analog voltage values had been collected, sorted from less to more, and then average had been taken for six center samples in Arduino IDE. The collected and average voltage value of them was finally converted into the pH value.

The pH value with corresponding port voltage had been displayed on the IDE terminal. The code is saved in IDE software as a sketch. In the tools tab of IDE software proper name of the board, "Arduino UNO", and port selection is a must before code upload.

Figure 12 shows the experimental setup of the pH measurement. First, the sensor is connected to the breakthrough board by a BNC connector. Next, the Vcc, ground, and P0 pin of the breakthrough board are connected to 5V, ground, and analog port pin A0 of the Arduino UNO. The Arduino UNO is then connected to the laptop by USB cable.

### *2.5. Sample Preparation*

All the infusions or tea samples were prepared by boiling tea with water in various proportions 100 mL, 170 mL, and 230 mL for five minutes, except green tea. The hot water had been poured into the green tea bag and kept for 5 min.The test mixtures were prepared without sugar, honey, and milk. Other additives such aslemongrass, Tulasi, ginger, and lemon are verified for the variations in the pH value of black tea (CTC) liquor. The liquor has been kept cool down to room temperature and then filtered by filter paper for each test.

### *2.6. Measurements*

#### 2.6.1. Mean

It is the center value of the selected numbers of output parameter samples. It is defined by the ratio of summations of all the values of selected output parameter samples to the total number of output samples. Equations (5) and (6) show the mathematical relation of the mean with sample value and the number of test samples.

$$\text{Mean (X)} = \frac{(X_1 + X_2 + \cdots + X_n)}{n} \tag{5}$$

$$\text{Mean (X)} = \sum \frac{X_i}{n} \tag{6}$$

where $X_i$ is the value of the ith test sample.
N = Total number of the test samples.

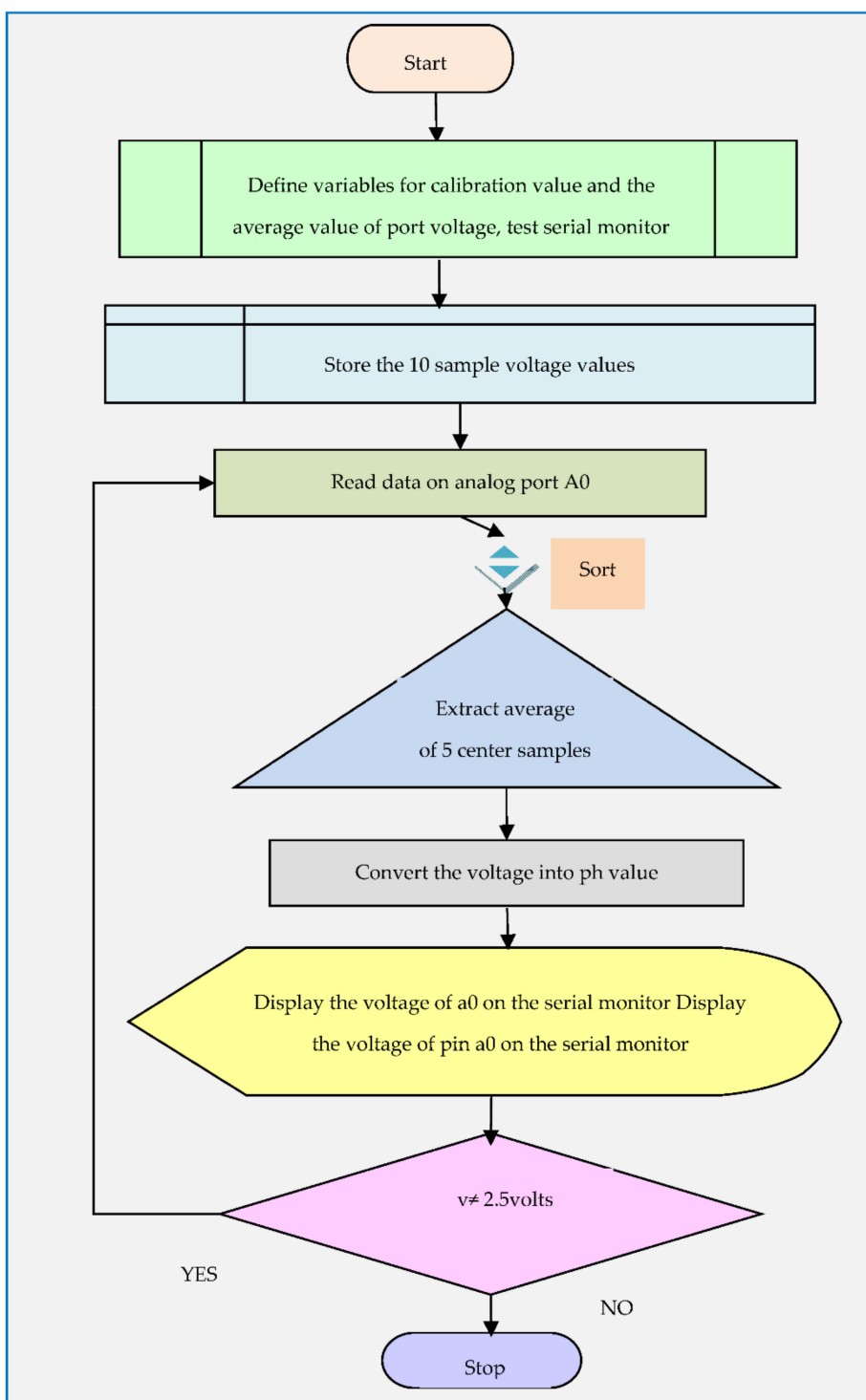

**Figure 11.** pH sensor measurement algorithm.

### 2.6.2. Standard Deviation

A standard deviation is useful to measure deviation from the mean value of the test sample. A standard deviation is defined as shown in "7". It is the square root of the average squares of deviations obtained from the arithmetic mean.

$$\text{Standard deviation } (\sigma) = \sqrt{\frac{\sum(Xi - X)^2}{n}} \tag{7}$$

Equations (5)–(7) had been used to analyze tea samples, and the result is stated in Table 8. In supervised machine learning, mean and standard deviations are useful measures.

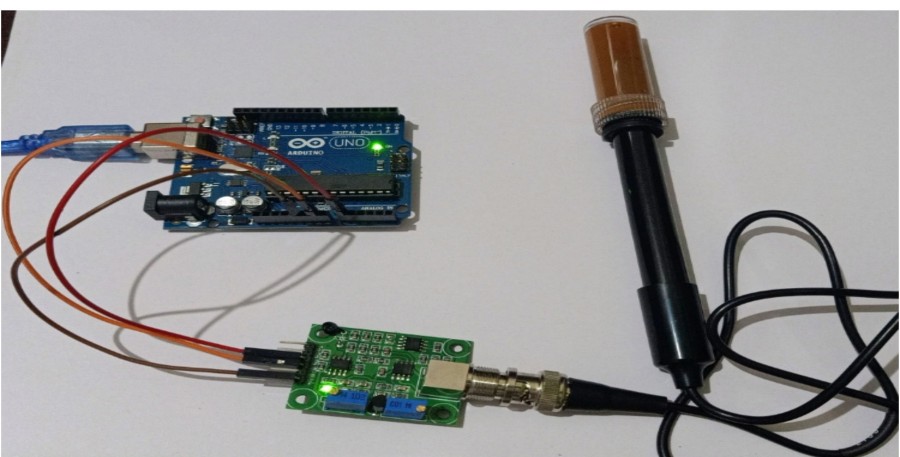

**Figure 12.** Experimental setup for pH measurement.

### 3. Results

*3.1. Research Data Description*

The observation table has been prepared under four test sets:

- Set 1–10 g, 5 min, 100 mL.
- Set 2–10 g, 5 min, 170 mL.
- Set 3–10 g, 5 min, 230 mL.
- Set 4–10 g, 5 min, 100 mL.

Set 1, 2, and 3 are formed for black tea, green tea, and energy mix analysis. The two types of black tea samples are under consideration-Orange pekoe and CTC. The duration of the sample preparation is the same. The effect on the pH values and port voltages had been observed and noted in Figures 13–15, respectively. Tables 4–6 indicate five sample values on the terminal for sets 1, 2, and 3, respectively. After each testing, the pH sensor had been washed with distilled water and then immersed into the next sample. Set 4 is specifically arranged for additives such asTulasi, lemongrass, ginger, lemon, and the temperature effect on the pH values of CTC. Figure 16 shows the picture of it. Table 7 shows a comparative analysis of all the liquors for set 4.

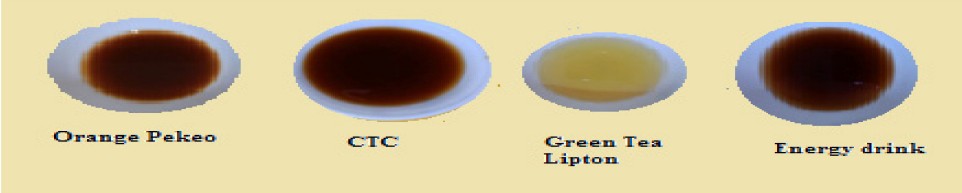

**Figure 13.** Tea infusion (Set 1–10 g, 5 min, 100 mL).

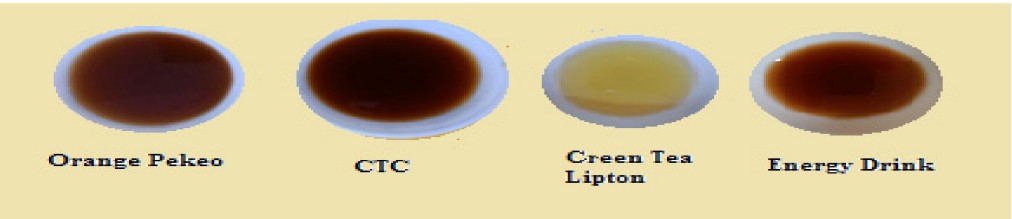

**Figure 14.** Tea infusion (Set 2–10 g, 5 min, 170 mL).

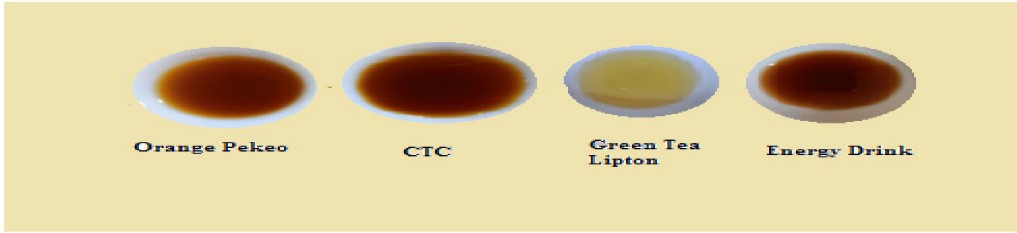

**Figure 15.** Tea infusion (Set 3–10 g, 5 min, 230 mL).

**Table 4.** pH value and the corresponding port voltage for Set 1–10 g, 5 min, 10 mL.

| Set 1–10 g, 5 min, 100 mL | | | | | | | |
|---|---|---|---|---|---|---|---|
| **Orange Pekoe** | | **CTC** | | **Green Tea (Lipton)** | | **Energy Drink Mix (Herbalife Nutrition)** | |
| **Voltage (V)** | **pH Value** | **Voltage (V)** | **pH Value** | **Voltage (V)** | **pH Value** | **Voltage (V)** | **pH Value** |
| 2.84 | 4.08 | 2.88 | 3.89 | 2.74 | 4.69 | 2.58 | 5.60 |
| 2.85 | 4.06 | 2.88 | 3.88 | 2.74 | 4.69 | 2.57 | 5.62 |
| 2.85 | 4.06 | 2.88 | 3.87 | 2.74 | 4.69 | 2.57 | 5.64 |
| 2.85 | 4.06 | 2.88 | 3.87 | 2.74 | 4.70 | 2.57 | 5.62 |
| 2.85 | 4.06 | 2.88 | 3.87 | 2.73 | 4.70 | 2.57 | 5.63 |

**Table 5.** pH value and the corresponding port voltage for Set 2–10 g, 5 min, 170 mL.

| Set 2–10 g, 5 min, 170 mL | | | | | | | |
|---|---|---|---|---|---|---|---|
| **Orange Pekoe** | | **CTC** | | **Green Tea (Lipton)** | | **Energy Drink Mix (Herbalife Nutrition)** | |
| **Voltage (V)** | **pH Value** | **Voltage (V)** | **pH Value** | **Voltage (V)** | **pH Value** | **Voltage (V)** | **pH Value** |
| 2.80 | 4.31 | 2.82 | 4.23 | 2.71 | 4.87 | 2.53 | 5.86 |
| 2.80 | 4.31 | 2.82 | 4.20 | 2.71 | 4.82 | 2.54 | 5.83 |
| 2.81 | 4.30 | 2.82 | 4.20 | 2.71 | 4.85 | 2.53 | 5.86 |
| 2.81 | 4.30 | 2.82 | 4.20 | 2.71 | 4.85 | 2.52 | 5.91 |
| 2.81 | 4.29 | 2.82 | 4.20 | 2.71 | 4.84 | 2.53 | 5.85 |

**Table 6.** pH value and the corresponding port voltage for Set 3–10 g, 5 min, 230 mL.

| Set 3–10 g, 5 min, 230 mL | | | | | | | |
|---|---|---|---|---|---|---|---|
| **Orange Pekoe** | | **CTC** | | **Green Tea (Lipton)** | | **Energy Drink Mix (Herbalife Nutrition)** | |
| **Voltage (V)** | **pH Value** | **Voltage (V)** | **pH Value** | **Voltage (V)** | **pH Value** | **Voltage (V)** | **pH value** |
| 2.77 | 4.48 | 2.76 | 4.50 | 2.67 | 5.06 | 2.48 | 6.14 |
| 2.77 | 4.48 | 2.76 | 4.51 | 2.67 | 5.05 | 2.49 | 6.11 |
| 2.77 | 4.48 | 2.76 | 4.48 | 2.67 | 5.04 | 2.49 | 6.12 |
| 2.77 | 4.48 | 2.76 | 4.49 | 2.68 | 5.04 | 2.49 | 6.10 |
| 2.77 | 4.48 | 2.76 | 4.48 | 2.68 | 5.00 | 2.49 | 6.12 |

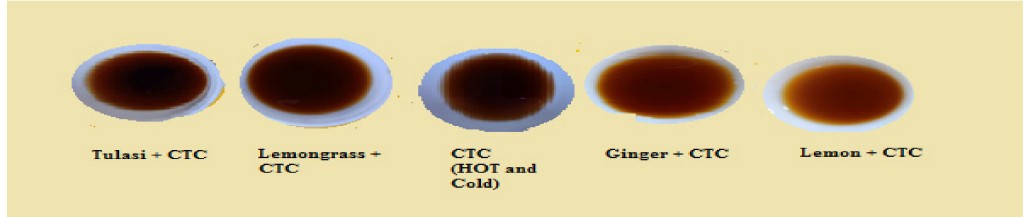

**Figure 16.** Tea infusion (Set 4–10 g, 5 min, 100 mL).

**Table 7.** pH value and the corresponding port voltage for Set 4–10 g, 5 min, 100 mL.

| Set 4–10 g, 5 min, 100 mL | | | | | | | | | | | |
|---|---|---|---|---|---|---|---|---|---|---|---|
| Tulasi + CTC | | Lemmon Grass + CTC | | CTC-HOT | | CTC-COLD | | Ginger + CTC | | Lemon + CTC | |
| Voltage (V) | pH Value | Voltage (V) | pH Value | Voltage (V) | pH Value | Voltage (V) | pH Value | Voltage (V) | pH Value | Voltage (V) | pH Value |
| 2.74 | 4.65 | 2.75 | 4.59 | 2.87 | 3.91 | 2.87 | 3.94 | 2.83 | 4.15 | 3.18 | 2.14 |
| 2.74 | 4.66 | 2.76 | 4.58 | 2.88 | 3.88 | 2.87 | 3.95 | 2.84 | 4.12 | 3.19 | 2.12 |
| 2.75 | 4.61 | 2.76 | 4.57 | 2.88 | 3.87 | 2.87 | 3.96 | 2.84 | 4.12 | 3.19 | 2.12 |
| 2.74 | 4.65 | 2.76 | 4.58 | 2.88 | 3.87 | 2.87 | 3.92 | 2.84 | 4.12 | 3.19 | 2.12 |
| 2.74 | 4.65 | 2.76 | 4.56 | 2.89 | 3.84 | 2.87 | 3.94 | 2.84 | 4.12 | 3.19 | 2.12 |

The tea sample mentioned in Table 6 has been prepared by boiling 10 g of tea in water with set 1 (2%), set 2 (6%), and set 3 (10%) proportions. The mean value and standard deviations for the pH value are discussed individually in Table 8.

**Table 8.** Comparative analysis of the tea samples with different mixture proportions for their pH values.

| Tea | Sample | Mixture Concentration | Mean (X) of the pH Value | Standard Deviation (σ) of the pH Value | Remark |
|---|---|---|---|---|---|
| Black Tea | Orange Pekoe | 2% | 4.065 | 0.00671 | Medium flavor (Less acidic) |
| | | 6% | 4.296 | 0.00800 | Medium flavor (Less acidic) |
| | | 10% | 4.48 | 0.00000 | Mild flavor (Safe Acidic) |
| | CTC | 2% | 3.869 | 0.01044 | Strong flavor (Acidic) |
| | | 6% | 4.196 | 0.01428 | Medium flavor (Less acidic) |
| | | 10% | 4.493 | 0.01100 | Mild flavor (Safe Acidic) |
| Green Tea | Lipton | 2% | 4.687 | 0.00781 | Mild flavor (Safe Acidic) |
| | | 6% | 4.83 | 0.02191 | Mild flavor (Safe Acidic) |
| | | 10% | 5.03 | 0.01844 | Mild flavor (Safe Acidic) |
| Energy Drink Mix | Herbalife Nutrition | 2% | 5.621 | 0.01640 | Mild refreshing flavor (Safe Acidic) |
| | | 6% | 5.879 | 0.02625 | Mild refreshing flavor (Safe Acidic) |
| | | 10% | 6.118 | 0.01077 | Mild refreshing flavor (Safe Acidic) |
| Tulasi Tea | CTC+Tulasi | 2% | 4.633 | 0.01952 | Mild flavor (Safe Acidic) |
| Ginger Tea | CTC+Ginger | 2% | 4.123 | 0.00900 | Medium flavor (Less acidic) |
| Lemongrass Tea | CTC+Lemon grass | 2% | 4.56 | 0.01789 | Mild flavor (Safe Acidic) |
| Black tea | CTC +HOT | 2% | 3.866 | 0.02107 | Strong flavor (Acidic) |
| Black tea | CTC+COLD | 2% | 3.937 | 0.01187 | Strong flavor (Acidic) |
| Lemon Tea | CTC+Lemon | 2% | 2.122 | 0.00600 | Very strong flavor (Acidic) |

Figures 17–19 are the graphs for the voltage versus pH plotting for set 1, set 2, and set 3 conditions, respectively. For all the sets, the pH value of the energy drink is comparatively high and depicts mild flavor with less acidity. The very next location of the plot is grabbed by green tea and is in 2nd position with a little bit acidic compared with an energy drink. Finally, the black tea types are overlapped on all the graphs shown in Figures 17–19 and depict comparatively higher pH and strong flavor for conditions of all three sets.

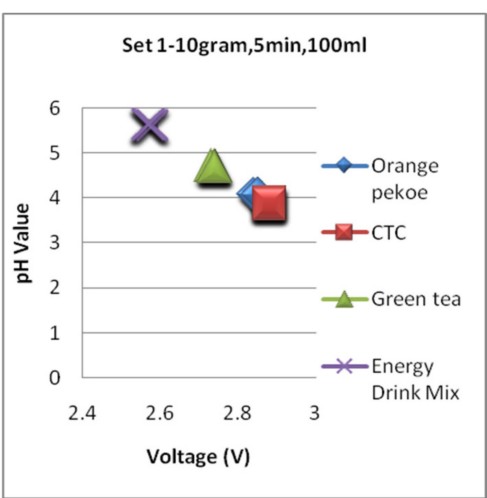

**Figure 17.** The voltagevs. pH plotting for Set 1.

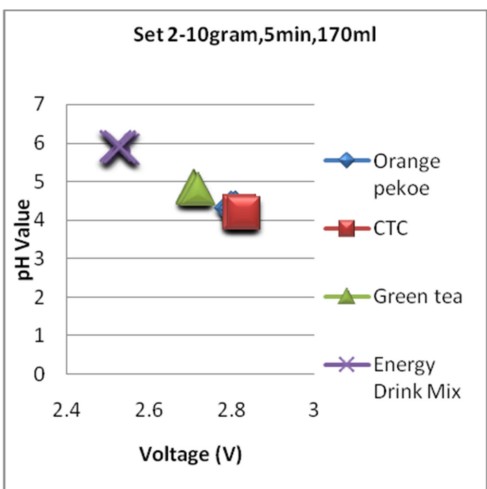

**Figure 18.** The voltagevs. pH plotting for Set 2.

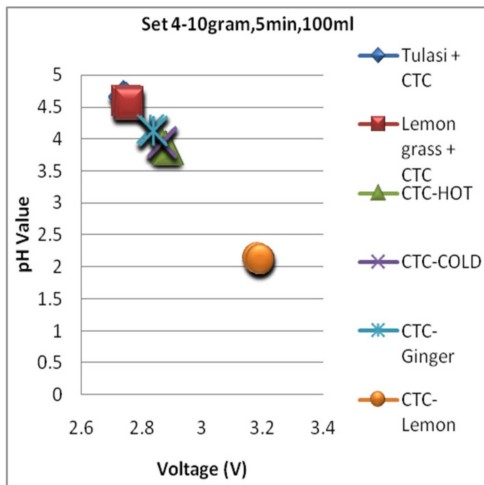

**Figure 19.** The voltagevs. pH plotting for Set 3.

Figure 20 shows the impact of the additives and temperature on CTC liquor. Ascending order of liquors for the pH value had been shown in Figure 20, where the CTC will become less acidic with the additives Tulasi, lemongrass, and ginger. In contrast, the acidity

will increase with the addition of lemon juice. In the case of temperature, the cold liquor is less acidic than the hot liquor.

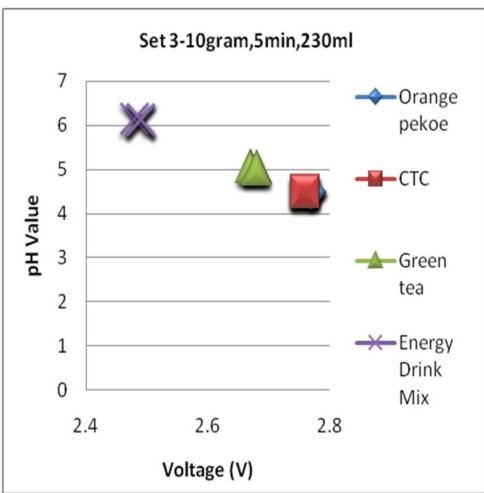

**Figure 20.** Impact of the additives and temperature on CTC pH value.

The ascending order of liquors according to its pH value is lemon + CTC < CTC HOT < CTC COLD < ginger + CTC < lemongrass + CTC < Tulasi + CTC.

### 3.2. Research Contributions

The standard dataset was not available for the various tea samples with their pH values in the open-access literature. The ranges of the pH values have been mentioned in the literature. The methods of infusion preparation and measurement mechanism have been compared in the literature. The impact of additives such as Tulasi, lemongrass, ginger, and lemon had not been found anywhere. Following are the research contributions of the paper:

- In this paper, for the same application conditions, the standard dataset of the pH value has been prepared and explained in detail, considering the two black tea samples, one green tea sample, and one energy drink sample.
- Detailed analysis of the impact of various tea additives such asTulasi, lemongrass, ginger, and lemonon the pH value of CTC has been conducted and explained in detail. The effect of temperature on the pH value of tea liquor had been analyzed, which was not elaborated in previous literature.

## 4. Discussion

An urge for human health awareness and safety leads to artificial taste perception. Sensors and software programming can be used to monitor results in beverage development, beverage purity confirmation, flavor ageing analysis in a liquid sample of beverages, alcoholic, or non-alcoholic drinks, and measures the effect of process control variables, establishing adherence to government standards, quantifies levels of spice, flavors, dissolved compounds, and quantifies taste-masking success.

In the food and beverage industries, predictive maintenance and personalized recommendation are very important. A brand's reputation may degrade due to product recall or being found to be contaminated, and that can be enough to ban a business as, throughitsuse, consumer's life maybe at risk. With the help of automation and technology, brands can be protected. It also helps in the reduction inwastage inlets, which will reduce end cost, production and maintenance time, and efforts.

The tea sample testing and grading results are mentioned in this paper. The flavor of the tea is dependent on its pH value and various additives. The temperature effect has

been discussed and it was found that as temperature raises the pH increases. Following are digital indicators of the tea sample testing by Sensory mechanism and Arduino Uno.

For the 2% concentration, the tea samples have been compared as given bellowCTC (pH value mean 3.869 and standard deviation 0.01044), Orange Pekoe (pH value mean 4.065 and standard deviation 0.00671), Lipton green tea (pH value mean 4.687 and standard deviation 0.00781), and Herbalife Nutrition (pH value mean 5.621 and standard deviation 0.01640). The flavor and color of the tea are directly associated with its pH value. As the pH value reduces, the color becomes darker, and the flavor becomes strong.

For the 2% concentration, the additives with base CTC sample have been compared as given bellow: lemon (pH value mean 2.122 and standard deviation 0.00600), ginger (pH value mean 4.123 and standard deviation 0.00900), lemongrass (pH value mean 4.56 and standard deviation 0.01789), and Tulasi (pH value mean 4.633 and standard deviation 0.01952). The CTC containinglemon is the most acidic in comparison with ginger, lemongrass, and Tulasi.

The impact of the temperature of the CTC tea liquor had been tested. For 2% concentration, CTC HOT liquor (pH value mean 3.866 and standard deviation 0.02107) and CTC COLD liquor (pH value mean 3.937 and standard deviation 0.01187) indicate a rise in temperature acidity of liquor increases, and deviation from mean is also high. Table 8 also indicates grading of the tea flavors for 6% and 10% concentrations of the mixture. The testing and analysis of tea given in the paper are only limited to pH value for the specific preparation time.

## 5. Conclusions

The early intimation is required on the consumer's side for the tea type, tea quantity, flavor, and preparation time. Today's environment of fast and long working hours leads to the consumption of tea. Teadrinking while consumingfood affectscalcium and nutrients absorption in the body and is not recommendedfor the age group 16–20 when physical activities are increased. Some special cases such aspregnancy and some diseases such as hyperacidity and heat problems allow for a limited consumption of tea. The paper describes the analysis of tea taste based on its pH value. Two black tea types—orange pekoe and CTC—were compared with green tea and one sample of energy drink with the same preparation conditions. The pH value is highly dependent on the liquor concentration proportions.

This paper contributes to the flavor grading for the tea liquor in terms of strong, medium, and mild taste.

**Author Contributions:** Conceptualization, M.B. and K.K.; methodology, A.P.; investigation, A.P.; resources, M.B., K.K. and A.P.; data curation, A.P.; writing—original draft preparation, A.P.; writing—review and editing, M.B. and K.K.; funding acquisition, M.B. and K.K. All authors have read and agreed to the published version of the manuscript.

**Funding:** This research was funded by "Research Support Fund of Symbiosis International (Deemed University)".

**Acknowledgments:** The authors would like to thank both SIU and SIT for their support.

**Conflicts of Interest:** The authors declare no conflict of interest.

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
