# Peer review of "Identification and Classification of the Tea Samples by Using Sensory Mechanism and Arduino UNO"

_inventions, doi:10.3390/inventions6040094_

Round 1

Reviewer 1 Report

Based on the title it appears authors wish to write about the application of machine learning for monitoring tea. However, the manuscript in this current state is very poorly written, disorganised and difficult to read. It is not even clear if it is a review paper of research article.  Why does the paper start with a summary instead of an introduction? The authors did not follow the MDPI guidelines for authors.

Other important comments:

I don’t see how the first paragraph of the summary (introduction), is in any way related to the objectives of the study.

Figure 1 is very difficult to understand. Is the “Virtual Demonstration of the Tea Flavor based on its pH value “or the “virtual demonstration of the tea flavor for its bitterness in terms of its pH value.”? I was a expecting the influence of different (pH scale) on the flavor or bitterness (Whichever one you are referring to).

Line 62: What do you mean by “The head alteration has set the benchmarks for standards and guidelines for such food processing”

Many strong claims were made from lines 50-94 but without any appropriate citation

There should be no subheadings in the introduction

Line 82: What do you mean by “The HPLC and GC equipment are around 8 to 10 lakhs” Please use standard units.

The aim of the study is missing in the introduction. There is also nothing about machine learning in the introduction.

Why does the materials and methods look like a literature review? Is this a review paper or a research article?

Author Response

The Response to the reviewer is submitted in the attached file

Point 1: Based on the title it appears the authors wish to write about the application of machine learning for monitoring tea. However, the manuscript in this current state is very poorly written, disorganized and difficult to read. It is not even clear if it is a review paper of a research article.  Why does the paper start with a summary instead of an introduction? The authors did not follow the MDPI guidelines for authors.

Response 1: Correction has been made for the introduction part; the paper has been started with the introduction instead of a summary. The guidelines and format given for the research article have been followed

Point 2: I don’t see how the first paragraph of the summary (introduction), is in any way related to the objectives of the study.

Response 2: The introduction is modified as per the suggestion of the reviewer and is made aligned with the objective of the study.

Point 3: Figure 1 is very difficult to understand. Is the “Virtual Demonstration of the Tea Flavor based on its pH value “or the “virtual demonstration of the tea flavor for its bitterness in terms of its pH value.”? I was expecting the influence of different (pH scale) on the flavor or bitterness (Whichever one you are referring to).

Response 3: The reviewer's suggestion has been accepted and accordingly the modification has been done in figure 1.

Point 4: Line 62: What do you mean by “The head alteration has set the benchmarks for standards and guidelines for such food processing”.

Response 4: In line 62, the Tea board of India has been mentioned as head, and it is the central authority that is providing standards for tea manufacturing. Still, to avoid confusion for the head term the sentence has been erased from the paper.

Point 5: Many strong claims were made from lines 50-94 but without any appropriate citation”

Response 5: Thank you for the suggestion. Reference number 67 has been included in paper references as a support document link for the claims made in lines 50-94. 

Point 6: There should be no subheadings in the introduction

Response 6: Thank you for the suggestion. The headings “Manual test and Analysis” and “Chemical test and Analysis” were mentioned under the introduction section of the paper have been removed now and no subheadings are there in the introduction part.

Point 7: Line 82: What do you mean by “The HPLC and GC equipment are around 8 to 10 lakhs” Please use standard units

Response 7: Thank you for the suggestion. The standard unit for equipment cost-Rupees has been mentioned as the cost opted is in Indian currency.

Point 8: The aim of the study is missing in the introduction. There is also nothing about machine learning in the introduction

Response 8: The points - need of the study, objective of the study, and utilization of the machine learning for the tea samples have been mentioned in the introduction part.

Point 9: Why do the materials and methods look like a literature review? Is this a review paper or a research article?

Response 9: The materials and methods have been separately mentioned in this paper revision and the literature review has been given after the introduction section as a background study to understand the topic importance and previous opted methods and work done in this domain.

Reviewer 2 Report

In this work, based on the supervised machine learning algorithm, tea samples of different levels are intelligently classified. This paper has obvious research significance. The experimental design and data processing process is clear and reasonable. Some revised opinions are as follows:
(1) Section 5.3, the main contribution of the paper is explained too detailed. As a research paper, 2-3 main contributions are enough, and please simplify.
(2) The conclusion has content too detailed and should not include the background. The conclusion should directly give the digital indicators and main results.
(3) The paper should be simplified, too detailed elaboration is not conducive to the reader's understanding. Some graphics and text can be placed in supporting materials.
(4) The author uses a supervised machine learning algorithm to classify tea grades intelligently. However, the algorithm modeling and classification process is less elaborated. It is not clear how the detection data is combined with the algorithm.

Author Response

The response to the reviewer comments is attached 

Round 2

Reviewer 1 Report

Currently, the paper is structured like a thesis.It still does not follow the template of the instructions for authors document on the invention websites: https://www.mdpi.com/journal/inventions/instructions 

There is no discussion.

The equations at the results section should be moved to materials and methods.

Author Response

Response to Reviewer 1 Comments

Point 1: Currently, the paper is structured like a thesis. It still does not follow the template of the instructions for the author’s document on the invention websites:  https://www.mdpi.com/journal/inventions/instructions 

Response 1: Correction has been made for all the sections of the paper as per the journal template suggested on https://www.mdpi.com/journal/inventions/instructions .

Point 2: There is no discussion.

Response 2: The discussion section had been given in this paper as suggested in the template guideline.

Point 3: The equations in the results section should be moved to materials and methods.

Response 3: The reviewer’s suggestion has been accepted and accordingly the modification has been done in the “results” and “materials and methods” sections.

Round 3

Reviewer 1 Report

The manuscript still does not follow the template of the instructions for the author’s document on the invention websites. E.g there should be no subsection for research problems and objectives in research articles. They should be merged in the introduction and the aims and objectives should be stated in the last paragraph of the introduction. 

The introduction, in the present is too long and makes the paper appear as a review paper. Same for the methods and conclusion. Authors need clarify if it is a review or research paper.